# Characteristics Associated with Improved Physical Performance among Community-Dwelling Older Adults in a Community-Based Falls Prevention Program

**DOI:** 10.3390/ijerph17072509

**Published:** 2020-04-06

**Authors:** Gabrielle Scronce, Wanqing Zhang, Matthew Lee Smith, Vicki Stemmons Mercer

**Affiliations:** 1Curriculum in Human Movement Science, University of North Carolina at Chapel Hill, Chapel Hill, NC 27599, USA; 2Department of Allied Health Sciences, University of North Carolina at Chapel Hill, Chapel Hill, NC 27599, USA; wanqing_zhang@med.unc.edu; 3Center for Population Health and Aging, Texas A&M University, College Station, TX 77843, USA; matthew.smith@tamu.edu; 4Department of Environmental and Occupational Health, School of Public Health, Texas A&M University, College Station, TX 77843, USA; 5Department of Health Promotion and Behavior, College of Public Health, The University of Georgia, Athens, GA 30602, USA; 6Division of Physical Therapy, University of North Carolina at Chapel Hill, Chapel Hill, NC 27599, USA; vicki_mercer@med.unc.edu

**Keywords:** older adults, falls, exercise

## Abstract

This was a retrospective analysis of quasi-longitudinal data from an ongoing, community-based falls prevention program. The purpose was to identify participant characteristics predicting improvement on physical performance measures associated with falls risk. Community-dwelling older adults ≥60 years old participated in a community-based implementation of the Otago Exercise Program (OEP). Participants with increased falls risk (*n* = 353) were provided with individualized exercises from OEP and were invited to return for monthly follow-up. One hundred twenty-eight participants returned for at least two follow-up visits within 6 months of their initial visit (mean time to second follow-up = 93 days with standard deviation = 43 days). Outcome measures assessed at initial and all follow-up visits included Four Stage Balance Test (4SBT), Timed Up and Go test (TUG), and Chair Rise Test (CRT). Distributions were examined, and results were categorized to depict improvement from initial visit (IVT) to second follow-up visit (F2). Key predictor variables were included in multivariable linear or logistic regression models. Improved 4SBT performance was predicted by greater balance confidence. Better TUG performance at F2 was predicted by no use of assistive device for walking, higher scores on cognitive screening, and better IVT TUG performance. Improvement on CRT was predicted by younger age and lower scores on cognitive screening. While improvements on each of the three measures were predicted by a unique combination of variables, these variables tended to be associated with less frailty.

## 1. Introduction

Falls are a common and often devastating occurrence among older adults. For older adults, falls can result in negative consequences such as physical injury, psychological distress, loss of independence, and death [1,2,3]. Fortunately, abundant research supports the efficacy of exercise-based interventions to reduce falls among community-dwelling older adults [4,5,6,7]. The Otago Exercise Program (OEP) is an example of an exercised-based intervention that has repeatedly resulted in significant reductions in falls among older adult participants in randomized controlled trials [8,9]. The OEP, originally designed and implemented in New Zealand, is a one-on-one home-based intervention consisting of 17 exercises to improve lower extremity strength and balance [10,11,12,13]. In the United States (US), licensed physical therapists (PTs) deliver OEP in a patient’s home, beginning with administering standardized tests to measure the patient’s balance and lower extremity strength. The PT individualizes OEP by selecting exercises that improve the patient’s balance and lower extremity strength, thereby reducing the patient’s risk for falls. In addition to this initial visit, the intervention includes follow-up visits to measure progress and advance exercises at 1, 2, 4, and 8 weeks after the initial visit [14,15,16]. Economic evaluations have found that the benefits of OEP are provided at a low program cost [17,18,19]. As a result of the fall prevention benefits and low costs associated with delivery, government and public health agencies have supported the use of OEP by clinicians in the US and abroad [20,21,22,23,24].

Despite the benefit, cost effectiveness, and support for OEP, adoption of the program by clinicians in the US has been limited [14]. As a result, alternative implementation strategies have been developed to increase OEP availability for older adults meeting certain characteristics and in specific geographic locations [25,26,27]. The Community Health and Mobility Partnership (CHAMP) was designed to offer OEP to communities with limited resources and/or underserved demographics, particularly those in rural areas without access to preventive care initiatives [28,29]. CHAMP is a creative, collaborative partnership, made up of academics, clinicians, health professions students, and community members who coordinate and deliver OEP in a central community location once per month from March to November. By providing the OEP in the community setting instead of in participants’ homes, CHAMP provides a greater number of individuals living in rural areas the opportunity to self-identify a concern about falls, receive assessment and intervention, and subsequently return for follow-up without undue access burden. 

CHAMP is delivered at no cost to participants because of time and resources donated by individual providers and volunteers, local clinics, and community centers. For programs like CHAMP that rely on volunteer efforts and limited resources, it is important to maximize program impact using the concepts of reach and effectiveness [30,31]. Impact can be optimized when a program reaches individuals who experience positive effects from participation [32]. Consequently, identification of those individuals most likely to improve physical performance with CHAMP participation is an important step for informing recruitment efforts and increasing the cost-efficiency of the program. The purpose of this study was to identify characteristics of CHAMP participants predicting improvement on physical performance measures associated with falls risk, including Four Stage Balance Test (4SBT) [33], Timed Up and Go test (TUG) [34], and Chair Rise Test (CRT) [35], and to compare these characteristics to those predicting improvement in the traditional home-based OEP.

Results from OEP delivered in New Zealand demonstrated that while all participants demonstrated improved strength and balance and a decreased number of falls, a subgroup of individuals aged at least 80 years old and with at least one fall in the previous year received the greatest benefit from participation based on estimated falls prevented [8]. Researchers concluded that the subgroup of older, frailer participants may benefit most from balance and strength gains provided by OEP to reduce falls risk, whereas younger individuals with better balance and strength at baseline have smaller gains [8]. Similar results and conclusions were found in implementations of OEP in the US [14]. Based on these findings, we hypothesized that characteristics of CHAMP participants predicting improvement would be those associated with increased frailty, such as increased age, greater number of falls, activity limitation because of fear of falling, use of an assistive device, decreased balance confidence, and cognitive impairment [8,36,37].

## 2. Materials and Methods 

### 2.1. Study Design and Participants

This study was a retrospective analysis of data collected from CHAMP over a span of 8 years. Participants were 128 community-dwelling older adults over the age of 60 years old who voluntarily participated in a CHAMP event and returned for two CHAMP follow-up visits within 6 months (182 days) after their initial visit (IVT). Mean time to second follow-up visit (F2) was 93 days with standard deviation (SD) = 43 days. F2 was utilized as the post-test visit because of previous research demonstrating improved physical performance after 8 weeks of participation in the US OEP [38]. With CHAMP offering one follow-up visit per month, two follow-up visits were the minimum needed to obtain a total treatment time of at least 8 weeks.

### 2.2. CHAMP

CHAMP participants were members of the community and surrounding areas who expressed interest in receiving free assessment of and intervention to reduce falls risk. All CHAMP events took place in a community senior or wellness center. Interested individuals were provided with a written description of program components and were asked to sign a consent form prior to participating. At the participant’s first visit, volunteer providers administered a multifactorial falls risk assessment that included tests used in OEP [12,13,38]. Based on test performance and other criteria described below, participants were determined to have either increased or low risk for falls. Those identified as having increased risk for falls were provided with a falls prevention intervention, which was based on the OEP and also included other evidence-based recommendations, such as use of grab bars in the bathroom. Under the direction of a licensed physical therapist, the OEP-based intervention involved CHAMP personnel selecting 3–5 exercises from OEP to meet each individual participant’s needs, training the participant in correct performance of the exercises, and providing the participant with written instructions for each exercise. When appropriate, participants were given ankle weights for home use. Participants were directed to perform strengthening exercises 3 days per week and balance exercises daily on their own at home. Participants were then advised to return for repeated monthly visits from March to November each year until their performance on falls risk measures improved sufficiently to indicate low risk for falls. At follow-up visits, falls risk measures were re-assessed, OEP exercises were reviewed and progressed, and additional falls risk factors were addressed as needed. For this study, we examined changes in performance on these measures from IVT to F2 to ensure participants had sufficient time to practice OEP home exercises provided by CHAMP.

### 2.3. Variables

#### 2.3.1. Baseline Participant Characteristics

Demographic and health history information was collected at each participant’s initial CHAMP visit. Participant information pertinent to this analysis included age, gender, race, education level, number and type of common chronic conditions, use of assistive device for walking, number of falls and injurious falls in previous year, and a yes or no response to the question “Do you limit your activities because you are afraid you might fall?” Balance confidence was measured by the Activities-specific Balance Confidence scale (ABC), a 16-item questionnaire in which the participant was asked to quantify percent confidence in maintaining balance during selected activities on an 11-point scale from 0 to 100 [39]. Items were averaged to determine overall percent confidence with higher scores indicating greater balance confidence [39]. The Mini-Mental State Examination (MMSE), which assesses orientation to time and place, registration and recall of three words, attention and calculation, language, and visual construction, was used for cognitive screening [40]. The MMSE was scored on a scale from 0 to 30 with a lower score reflecting greater cognitive impairment [40,41]. Body mass index (BMI) was calculated using participant’s measured body weight and reported height. 

#### 2.3.2. Outcome Measures

Three measures, 4SBT [33], TUG [34], and CRT [35], were assessed at each CHAMP visit to quantify an individual’s falls risk related to physical performance [13]. The 4SBT is a measure of static standing balance in each of four stance positions (feet together, semi-tandem, tandem, and single limb stance) for up to 10 seconds each [13,42,43]. The 4SBT score was the sum of the time (up to 10 seconds) that a participant was able to maintain each of the four positions without upper extremity support or loss of balance. Possible scores ranged from 0 to 40.0, with higher scores indicating better balance. A score of 35.0 or lower suggested that a participant was unable to maintain the single limb stance position for at least 5 seconds, a finding that has been associated with an increased risk for falls [44]. 

The TUG measured dynamic balance and functional mobility as the time needed to rise from a standard armchair, walk a distance of 3 meters, turn around, walk 3 meters back to the chair, and sit down in the chair [34]. Previous research has found that a TUG score of ≥12 seconds is associated with increased falls among older adults and that a minimal clinically important difference (MCID) ranges from 0.8 to 1.4 seconds among older adults with hip osteoarthritis [42,45,46,47,48]. At CHAMP, participants performed three trials of the TUG, one practice trial and two test trials, and results of the two test trials were averaged to produce the TUG score. 

Functional lower extremity strength was measured using CRT, a test of repeated sit-to-stand transfers performed independently and without upper extremity support. From 2009 to 2014, CHAMP providers administered the Chair Rise Test (CRT) as the 5 Times Sit-to-Stand (5xSTS), which has an MCID of 2.3 seconds, [49] in accordance with OEP instructions [13]. In 2015, CRT methods were changed from 5xSTS to the 30 second Sit-to-Stand test (30-s STS) to match recommendations from the Centers for Disease Control and Prevention [42,50]. For these analyses, a common CRT variable was computed by dividing the number of stands by time to produce a value with units of number of stands per second.

### 2.4. Statistical Analyses

Descriptive statistics were calculated for all variables. Frequencies and distributions were assessed for baseline characteristics considered for evaluation as predictor variables based on previous literature. Predictor variables with skewed distributions were dichotomized. 

Paired-samples t-tests were performed to assess change in scores on physical performance measures 4SBT, TUG, and CRT from IVT to F2. Effect sizes were calculated using Cohen’s d with common conventions used to classify values [51]. Furthermore, 4SBT and CRT were dichotomized to represent any improvement from IVT to F2 and no change or a worse score at F2 compared with IVT. 

Bivariate and multiple regression analyses were used to examine potential predictors for changes in outcome measures. Potential multicollinearity among the independent variables was examined prior to the multivariable analysis. Initial univariate analyses with baseline characteristics selected as predictor variables were carried out by logistic regression modeling improvement compared with no improvement or linear regression modeling F2 score as the dependent variable. A limited number of predictor variables were selected based on statistical significance of *p* < 0.1 on univariate regression analysis. Multivariable linear regression analysis was performed to identify associations between TUG and predictor variables, while logistic regression models were applied for 4SBT and CRT. In all models, predictor variables included age, use of assistive device for walking (yes/no), ABC, MMSE, and limited activity because of fear of falling (yes/no). Data were analyzed using SAS 9.4 (SAS Institute Inc., Cary, NC, USA). The significance level was set at *p* < 0.05 for all analyses. 

## 3. Results

### 3.1. Participant Characteristics

A total of 483 community-dwelling adults participated in at least one CHAMP event, with 28 excluded from data analysis because of incomplete initial visit or age <60 years. Of these, 353 individuals were recommended to return for CHAMP follow-up, with 117 (33.1%) not returning. Of the 236 participants who received follow-up, 151 returned for two CHAMP follow-up visits. To reduce confounding effects of time on change in performance, this study limited inclusion to 130 individuals whose return to F2 occurred within 6 months of their IVT. Out of these 130 participants, 128 were eligible for inclusion based on return to F2 within 6 months of F2 as well as having IVT and F2 visit scores recorded for 4SBT, TUG, and CRT. 

Baseline characteristics of these 128 participants are shown in Table 1. Mean age of CHAMP participants was 76.1 years (SD 8.1). Three quarters were female, 16.4% used an assistive device for ambulation, 57.5% reported at least one fall in the previous year, and 58.3% reported that they limited activity because of fear of falling.

Table 2 provides information about mean changes in 4SBT, TUG, and CRT scores from IVT to F2. When the physical performance measures were each dichotomized to distinguish CHAMP participants who improved (any positive change, regardless of magnitude) on the measure from those who stayed the same or got worse, improvements were seen for 76 (59.8%), 73 (57.0%), and 69 (60.5%) participants on the 4SBT, TUG, and CRT, respectively. 

### 3.2. Characteristics Associated with Improvement

In single-predictor logistic regressions modeling improvement in 4SBT, greater balance confidence measured by baseline ABC score was the only significant predictor. In univariable linear regressions modeling TUG score at F2, greater ABC, no use of assistive device, no falls in past year, higher MMSE, and better initial visit TUG were significant predictors of better TUG performance. Younger age was a significant single predictor in univariable logistic regression modeling improvement in CRT. Based on these univariable analyses, multivariable models were examined that included ABC, age, device, fall, and MMSE as predictor variables. 

#### 3.2.1. SBT

Multiple variable logistic regression showed that the only significant predictor of improvement in 4SBT was greater balance confidence measured by ABC at the initial visit. Table 3 shows the results of the logistic regression modeling improvement in 4SBT.

#### 3.2.2. TUG

Table 4 shows the results of a multiple variable linear regression modeling F2 TUG. Lower (better) F2 TUG scores were significantly predicted by no use of assistive device, higher scores on MMSE, and lower (better) IVT TUG score. 

#### 3.2.3. CRT

Table 5 depicts the results of a logistic regression modeling improvement on CRT. According to the model, improved CRT was predicted by younger age and higher scores on MMSE at IVT.

## 4. Discussion

Results from this study did not support our hypothesis that individuals with characteristics associated with frailty would demonstrate improvement on measures of physical performance from CHAMP IVT to F2. In fact, while significant predictors were different for each of the three physical outcome measures examined as dependent variables, improvement was consistently observed for individuals without characteristics associated with frailty across the three models. 

Surprisingly, initial univariate analyses revealed no significant associations of gender, number or type of comorbidities, or activity limitation with 4SBT, TUG, or CRT. The distribution of variables for gender, number of comorbidities, and presence of individual types of comorbidities was skewed among our sample and therefore may have increased the potential for Type II error. Distribution for the variable “limit,” however, was roughly equal, with 58.3% of participants reporting that they limited their activity because of fear of falling. The lack of significance associated with this variable and positive or negative change in 4SBT, TUG, or CRT may suggest that the single yes/no question related to activity limitation does not sufficiently capture the complex relationships among psychological factors and physical performance measures. It is possible that a more detailed assessment of activity limitation, such as the Survey of Activities and Fear of Falling in the Elderly (SAFE) [52], could be a stronger predictor of change in physical performance than the question “Do you limit your activities because you are afraid you might fall?” [52,53,54].

Although history of at least one fall in the previous year was a significant independent predictor of TUG performance at F2, fall history was not a significant predictor of improvement in 4SBT or CRT or of F2 TUG performance in multivariable models. Though contrary to our hypothesis that positive history of falls would predict improvement, the absence of significant association between fall history and physical performance from IVT to F2 suggests that improvement in physical performance can occur for participants regardless of fall history. 

In the multivariable model including age, use of an assistive device, fall history, and MMSE, balance confidence measured by ABC was the only significant predictor of improvement in 4SBT from initial visit to F2. There has been limited research investigating the psychometric properties of the 4SBT as a single test, but it is possible that the strong correlation between ABC and 4SBT overshadowed potential contributions from other variables included in this model predicting improvement in 4SBT. 

Our multivariable model showed that TUG performance at F2 was better for individuals with higher balance confidence measured by ABC, no use of assistive device, greater cognitive performance measured by MMSE, and better performance on TUG at IVT. In other words, participants with better baseline performance were more likely to have higher TUG scores at F2. 

Improvement in CRT was significantly predicted by younger age and lower MMSE in the multivariable model that also included ABC, use of device, and fall history. That younger CHAMP participants have greater odds of improving on the CRT is consistent with our other findings that characteristics not associated with frailty predict improvement on 4SBT and better scores on F2 TUG. Improvement in CRT for individuals with lower MMSE may be related to the ease of the CRT test and of the chair rise exercise from OEP, which is often prescribed to CHAMP participants. The chair rise exercise is a straightforward exercise that can be practiced without additional equipment and that participants are typically comfortable performing on their own. Individuals who practice the chair rise exercise are likely to improve their performance on the CRT because the exercise is so similar to the test. Based on the combination of benefits from task-specific practice and the simplicity of the test itself, the CRT may be a better measure of change in physical performance for individuals with cognitive impairment than the TUG or 4SBT [55].

The original OEP designed in New Zealand and delivered in an older adult’s home was most effective for individuals over 80 years of age and with increased frailty [8]. While specific relationships varied by each outcome measure, we found that CHAMP participants most likely to demonstrate improvement in physical performance tests were younger and without characteristics commonly associated with frailty. The differences in characteristics predicting improvement may be related to differences in the settings in which CHAMP and traditional OEP have been provided. Traditional OEP can be delivered to older adults who are unable to leave their homes, whereas CHAMP requires participation in a community setting. A result of the difference in settings is that successful participation in CHAMP requires greater community mobility than traditional OEP. As a result, individuals likely to maintain participation in CHAMP from IVT to F2 and achieve benefits from the program may be younger and less frail than participants who typically experience the greatest benefit from traditional OEP [12,56,57]. CHAMP providers can use this information to identify participants with increased age and frailty who require additional support to maximize their level of improvement on physical performance with CHAMP participation.

## 5. Limitations

Participants included in this analysis were restricted to those who elected and were able to attend a falls screening and intervention in the community, were identified as having an increased risk for falls, and followed recommendations to return for two subsequent visits in less than 6 months. These inclusion requirements may have resulted in selection bias that influenced differences in participant characteristics between those who improved in physical performance with CHAMP participation compared with traditional OEP participation. Other aspects of selection bias, such as the very limited inclusion of men, may also have affected our results. While our sample was representative of the older adult population of western North Carolina, the homogeneity of gender and race limits applicability of our findings to other geographical areas. 

Another limitation in this study was participant attrition. Of the 353 individuals who completed their initial visit to CHAMP and were advised to return for follow-up, 238 individuals returned only once and 128 returned for at least two follow-up visits. Based on the available data, there were very few baseline differences between individuals who returned compared with those who did not return for follow-up. While this level of attrition is expected in community-based programs targeting older adults compared with research studies that offer additional incentives such as monetary benefits, future assessment of CHAMP could include additional contacts with individuals who did not return to identify and address reasons why participants did not continue with the program. Measures included in this analysis were limited to those that are quick and easy to implement in a community setting by a variety of different providers with a variety of experience. The use of additional measures, such as those that could include other risk factors for falls, track adherence to program exercises and other recommendations, and record a count of falls, over a longer period of time with more regular follow-up would assist in clarifying the role of participant characteristics in predicting improvement with CHAMP participation.

## 6. Conclusions

This study found that significant predictors of improvement on physical performance measures vary depending upon the measure, but that individuals who benefit from a community-based falls prevention program that includes OEP are generally those with younger age, higher balance confidence, no need for an assistive device, and better baseline performance. 

## Figures and Tables

**Table 1 ijerph-17-02509-t001:** Baseline Characteristics of Participants (*n* = 128).

Characteristic, Units (Valid *n*)	Mean ± SD or *n* (%)
**Age**, *years* (*n = 128*)	76.1 ± 8.1
**BMI**, *kg/m^2^* (*n = 118*)	29.4 ± 6.6
**ABC**, *% confident* (*n = 128*)	63.3 **±** 19.7
**Number of chronic conditions** (*n* = 128)	2.9 ± 1.5
**Presence or history of common chronic conditions**	
Arthritis (*n = 127*) Hypertension (*n = 127*) Obesity (*n = 118*) Diabetes mellitus (*n = 126*) Cardiovascular disease (*n = 127*) Cancer (except skin cancer) (*n = 127*) Osteoporosis (*n = 127*) Stroke (*n = 127*)	94 (74.0%) 90 (70.9%) 46 (39.0%) 34 (27.0%) 31 (24.4%) 30 (23.6%) 26 (20.5%) 18 (14.2%)
**Falls in past year** (*n = 127*)	
Number of falls 1 or more	1.5 ± 3.2 73 (57.5%)
**Falls causing injury in past year** (*n = 127*)	
Number of falls 1 or more	0.3 ± 0.6 26 (20.5%)
**MMSE** (*n = 123*)	27.6 ± 3.1
**Gender**, *women* (*n = 127*)	96 (75.0%)
**Race** (*n = 120*)	
White African American Prefer not to answer	116 (96.7%) 3 (2.5%) 1 (0.8%)
**Education** (*n = 119*)	
< HS diploma or GED HS diploma or GED Associate’s degree Bachelor’s degree Graduate degree	12 (10.1%) 54 (45.4%) 21 (17.7%) 16 (13.4%) 16 (13.4%)
**Limits activity out of fear of falling** (*n = 127*)	74 (58.3%)
**Walks with assistive device** (*n = 128*)	
Yes Straight cane Quad cane Walker No	21 (16.4%) 11 (8.6%) 3 (2.3%) 7 (5.5%) 107 (83.6%)

Mean ± SD provided for continuous and *n* (%) for categorical data. Percentages exclude missing data. Abbreviations: *n*, sample size; valid *n*, number of participants with valid, non-missing data; SD, standard deviation; BMI, body mass index; ABC, Activities-specific Balance Confidence scale; MMSE, Mini-Mental State Examination; HS, high school; GED, general education degree; F2, Follow-up Visit 2; IVT, initial visit.

**Table 2 ijerph-17-02509-t002:** Mean Changes in Physical Performance Measures.

	IVT Score Mean ± SD	F2 Score Mean ± SD	Mean Improvement	*n*	*t*	*p*-Value	Cohen’s d
4SBT	29.5 ± 6.6	31.5 ± 7.2	2.0	127	−3.537	0.001	0.291
TUG	12.7 ± 5.5	11.9 ± 5.0	0.8	128	2.346	0.021	0.153
CRT	0.258 ± 0.132	0.290 ± 0.137	0.032	114	−3.233	0.002	0.239

Abbreviations: IVT, initial visit; SD, standard deviation; F2, follow-up visit 2; *n*, number of participants with valid, non-missing data; 4SBT, Four-Stage Balance test with units seconds; TUG, Timed Up and Go test with units seconds; CRT, Chair Rise Test with units stands per second.

**Table 3 ijerph-17-02509-t003:** Logistic regression analysis of improvement on 4SBT from Initial Visit to Follow-up Visit 2.

Predictor	*p*-Value	Odds Ratio	95% CI
ABC	0.028	1.025	(1.003, 1.047)
Age	0.965	0.999	(0.951, 1.049)
Device (Yes)	0.816	1.142	(0.372, 3.504)
Fall (Yes)	0.759	0.75l7	(0.513, 2.496)
MMSE	0.472	1.050	(0.920, 1.198)

Abbreviations: 4SBT, Four-Stage Balance Test; SE, standard error; df, degrees of freedom; CI, confidence intervals; ABC, Activities-specific Balance Confidence scale; Device, use of assistive device for walking with 1 = yes and 0 = no; Fall, occurrence of fall in year prior to IVT with 1 = 1 or more falls and 0 = 0 falls; MMSE, Mini-Mental State Examination.*.

**Table 4 ijerph-17-02509-t004:** Linear regression analysis modeling TUG performance at Follow-up Visit 2.

Predictor	Estimate	SE	*p*-Value
ABC	−0.026	0.019	0.163
Age	0.019	0.038	0.620
Device (yes)	2.470	1.000	0.015
Fall (yes)	0.797	0.627	0.206
MMSE	−0.324	0.103	0.002
IVT TUG	0.427	0.075	<0.001

Abbreviations: TUG, Timed Up and Go test; SE, standard error; df, degrees of freedom; CI, confidence intervals; ABC, Activities-specific Balance Confidence scale; Device, use of assistive device for walking with 1 = yes and 0 = no; Fall, occurrence of fall in year prior to IVT with 1 = 1 or more falls and 0 = 0 falls; MMSE, Mini-Mental State Examination; IVT TUG, Initial Visit Timed Up and Go test.

**Table 5 ijerph-17-02509-t005:** Logistic regression analysis of improvement on CRT from Initial Visit to Follow-up Visit 2.

Predictor	*p*-Value	Odds Ratio	95% CI
ABC	0.452	1.009	(0.986, 1.032)
Age	0.004	0.918	(0.867, 0.972)
Device	0.332	1.915	(0.515, 7.117)
Fall	0.825	0.909	(0.389, 2.214)
MMSE	0.023	0.795	(0.652, 0.968)

Abbreviations: CRT, Chair Rise Test; SE, standard error; df, degrees of freedom; CI, confidence intervals; ABC, Activities-specific Balance Confidence scale; Device, use of assistive device for walking with 1 = yes and 0 = no; Fall, occurrence of fall in year prior to IVT with 1 = 1 or more falls and 0 = 0 falls; MMSE, Mini-Mental State Examination.

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
