# Peer review of "Characteristics Associated with Improved Physical Performance among Community-Dwelling Older Adults in a Community-Based Falls Prevention Program"

_ijerph, 2020, doi:10.3390/ijerph17072509_

Round 1
Reviewer 1 Report
The current study assesses the predictive value of baseline characteristics of community-dwelling older participants for positive outcomes of a falls preventive single intervention program on intermediate mobility outcomes, eg TUGT, 4SBT and CST. The program that is assessed is the OTAGO exercise program and makes use of observational data. And specifically a free-of-charge version for communities with limited resources and/or underserved areas is assessed.
Abstract: include information on number of participants and drop-out rate; please state clearly time to follow-up median or mean and SD/quartiles.
Introduction: relevance and aim are clear, however the population under study is described already with very much detail. Please summarize for the introduction part and transfer details to methods section
Methods:
-please include METC and patient informed consent documentation in the methods description
-unclear what the timing of F2, visit two is. Please state clearly: given monthly visits was it the visit at two months follow-up/intervention? Or, after 6 months as mentioned in the first sentence of methods section? Or did it vary and if so, what was the distribution?
-Why were patients with one follow-up visit –within 6 months- not eligible for inclusion? Since only one follow-up visit was assessed, this does not appear to be logical or needed
-The paper would improve greatly with regard to relevance and comparability if there would be to have information on the actual outcome fall incidents during follow-up.
-statistical analyses: for determination of the full model, P 0.1 is quite strict, if p0.2 was taken as a cut of, does that influence the final model?
Statistical analyses: although the aim is to predict a positive outcome, eg effectiveness of the intervention on different mobility measures, the authors have confined themselves to multivariate regression models. In order to get a proper view on the predictive value of the (combined) baseline variables further steps need to be taken, ao calculation of the AUC
Results:
Although the mean differences of the different tests are shown, clinical relevance of these differences remains unclear. Please add, preferably in methods section and/or introduction information on the minimal clinical relevant difference
Discussion:
Please remove all causal explanations as prediction models cannot be translated into explanatory/causal relationships. Confine to strengths and weaknesses of the predictive value of your models.
Page 7, row 247: add mobility after improvement. Throughout the discussion section it needs to remain clear that predictive variables for improvement of mobility outcomes were assessed.
Page 9, line 309:the current study does not give information to better understand performance measures associated with fall risk. Fall risk was not an outcome in the current analyses. Please remove and rephrase
Please add to the discussion section information on the clinical relevance of the observed changes/improvement of the different outcomes. Were they clinically relevant?
Limitations section: given its observational nature and large drop out the study has serious limitations, that are currently only summed up shortly. Please add the possible consequences of the different limitations/biases to your limitation section. And also use it when referring to differences with current literature. For example potential bias introduced by difficulties for frailer groups to adhere to the CHAMP program are very likely to majorly explain the differences between the current study and other publications. Although mentioned, this needs further elaboration.
Please discuss to possible effects of the large drop out, describe non-participants and how this bias might have affected the findings.
Conclusion: predictive variables for improvement of mobility measures cannot be blindly assumed to also predict decreased fall risk. Please rephrase conclusion.
In the conclusion, the authors have turned around the research aim and translate it to identification of individuals who would need a different approach, with more support etc. Although very interesting and clinically very relevant, this was not assessed in the current study, so please rephrase more cautiously.
Author Response
Manuscript ID: ijerph-741189
Status: Pending major revisions
Characteristics Associated with Improved Physical Performance among Community-dwelling Older Adults in a Community-based Falls Prevention Program
International Journal of Environmental Research and Public Health
REVIEWER #1:
The current study assesses the predictive value of baseline characteristics of community-dwelling older participants for positive outcomes of a falls preventive single intervention program on intermediate mobility outcomes, eg TUGT, 4SBT and CST. The program that is assessed is the OTAGO exercise program and makes use of observational data. And specifically a free-of-charge version for communities with limited resources and/or underserved areas is assessed.
Abstract: include information on number of participants and drop-out rate; please state clearly time to follow-up median or mean and SD/quartiles.
- In Line 21, we added the number of people with increased falls risk. In Line 22, we added the number of people who returned for at least 2 follow-up visits within 6 months of their initial visit. In Lines 23-24, we added the mean time to second follow-up.
Introduction: relevance and aim are clear, however the population under study is described already with very much detail. Please summarize for the introduction part and transfer details to methods section
- We appreciate this reviewer’s comment. In the Introduction, we restricted the population description to include what is needed to justify and explain the study purpose and hypotheses. Specific details about the study participants are provided in the methods and results sections.
Methods:
-please include METC and patient informed consent documentation in the methods description
- In Lines 103-105, we added the consent form.
-unclear what the timing of F2, visit two is. Please state clearly: given monthly visits was it the visit at two months follow-up/intervention? Or, after 6 months as mentioned in the first sentence of methods section? Or did it vary and if so, what was the distribution?
- In Lines 96-97, we added the mean time to F2.
-Why were patients with one follow-up visit –within 6 months- not eligible for inclusion? Since only one follow-up visit was assessed, this does not appear to be logical or needed
- Given the aim of this study, only those with data at the three designated time points were included. We provided a justification for this inclusion criteria, which is justified by other previous effectiveness studies using a variation of this intervention (see lines 97-100).
-The paper would improve greatly with regard to relevance and comparability if there would be to have information on the actual outcome fall incidents during follow-up.
- We have noted this possible limitation on line 348 of the revised manuscript.
-statistical analyses: for determination of the full model, P 0.1 is quite strict, if p0.2 was taken as a cut of, does that influence the final model?
- We appreciate this recommendation. We have reviewed this suggestion. Applying a p-value <.2 does not change the model compared to a p-value of <.1.
Statistical analyses: although the aim is to predict a positive outcome, eg effectiveness of the intervention on different mobility measures, the authors have confined themselves to multivariate regression models. In order to get a proper view on the predictive value of the (combined) baseline variables further steps need to be taken, ao calculation of the AUC
- We appreciate this reviewer’s suggestion. We did not calculate AUC because studies have previously established recommended clinical thresholds for most of our outcome variables/physical performance indicators. Therefore, as an alternative, sensitivity tests were conducted including all potential predictors in the stepwise regression analyses based on clinical significance to corroborate our findings. As a team, we assessed and deliberated about all options and respectfully maintain our original analysis plan and methodology.
Results:
Although the mean differences of the different tests are shown, clinical relevance of these differences remains unclear. Please add, preferably in methods section and/or introduction information on the minimal clinical relevant difference
- We appreciate this recommendation. We have added the minimal clinically important difference (MCID) for TUG on Lines 150-151. Further, we added the MCID for CRT on Lines 156-157.
Discussion:
Please remove all causal explanations as prediction models cannot be translated into explanatory/causal relationships.
- We agree with this reviewer’s recommendation and have removed lines 273-285.
Confine to strengths and weaknesses of the predictive value of your models.
- We have changed Lines 270-285 to reflect this comment.
Page 7, row 247: add mobility after improvement. Throughout the discussion section it needs to remain clear that predictive variables for improvement of mobility outcomes were assessed.
- These suggested edits were made on Lines 266 and 312.
Page 9, line 309:the current study does not give information to better understand performance measures associated with fall risk. Fall risk was not an outcome in the current analyses. Please remove and rephrase
- This phrasing was removed (see lines 334-335).
Please add to the discussion section information on the clinical relevance of the observed changes/improvement of the different outcomes. Were they clinically relevant?
- We thank the reviewer for this question. The clinical relevance of the observed changes was beyond the scope of this paper; however, such an examination is a future direction of needed research.
Limitations section: given its observational nature and large drop out the study has serious limitations, that are currently only summed up shortly. Please add the possible consequences of the different limitations/biases to your limitation section. And also use it when referring to differences with current literature. For example potential bias introduced by difficulties for frailer groups to adhere to the CHAMP program are very likely to majorly explain the differences between the current study and other publications. Although mentioned, this needs further elaboration.
- In response to this comment, we have expanded upon and elaborated about the potential study limitations and their impact on the interpretation of findings (see Lines 327-329).
Please discuss to possible effects of the large drop out, describe non-participants and how this bias might have affected the findings.
- In response to this comment, we have expanded upon and elaborated about attrition as a potential study limitation (see Lines 337-344).
Conclusion: predictive variables for improvement of mobility measures cannot be blindly assumed to also predict decreased fall risk. Please rephrase conclusion.
- The physical performance measures used in this study have been shown to have strong associations with falls and are recommended by the CDC for use in screening for falls risk. We have rephrased the conclusion (see Lines 354-358).
In the conclusion, the authors have turned around the research aim and translate it to identification of individuals who would need a different approach, with more support etc. Although very interesting and clinically very relevant, this was not assessed in the current study, so please rephrase more cautiously.
- We appreciate this review. We agree with this assessment and have modified the conclusion to reflect this comment (see Lines 354-358).
Reviewer 2 Report
- This study is a retrospective/secondary analysis of existing data from the CHAMP. This study should be improved in one of two ways because current one is not answering an important question.
In the introduction, just before the purpose statements, it is written that “Additionally, understanding characteristics of participants who are less likely to demonstrate improvements may assist providers with decision-making when considering individuals’ need for referral to other resources”; therefore, I expected that the study would find who did not improve. But the study purpose was to find who improved. The author should find out who did not improve in this study, only then the current conclusion can be stated. The conclusion was not directly delivered from this study. The statistical analysis should be the same for logistic regression, except the authors need to discuss who did not improve.
If that is not what the authors want to do, it should be stated upfront that the CHAMP used OEP in a community-setting and the purpose is to compare the OEP’s home-based results in the original study and CHAMP’s community-based results. In this way, the authors can use almost all parts of the manuscript except for the rational (the reasons why this study should be conducted) for the study.
The following suggestions may be still relevant after revising the major issue above.
- Add the number of data that was analyzed (128?) in the abstract.
- Clarify the duration of Otago Exercise program per person (6 months?) in the abstract.
- Explain the setting of assessment and Otago training and what instruction for home-based exercise was provided (community-setting?).
- On page 4, line 158, CRT should be TUG. But the results of TUG should be categorized to improved and not-improved and use logistic regression.
- Lines 264-268 is difficult to understand because of the double negative word “decreased characteristics of frailty.” The same request for the conclusion.
- Lines 264-274. Why this paragraph is related to hypothesis or research question (RQ)? If it is, state the related hypothesis or RQ.
- The rest is good. It all made sense when I read Lines 287-300, but what it means is that until that paragraph, I was not sure why the authors are not answering the important questions.
- Explain why OEP was adapted for a community-setting in CHAMP, when it was meant for home-based exercise intended for those whom tend to be home-bound? Older adults who can come to the community-setting is different from older adults who tend to be home-bound.
- In CHAMP, explain what was provided for home OEP. No all readers know details of CHAMP.
Author Response
Manuscript ID: ijerph-741189
Status: Pending major revisions
Characteristics Associated with Improved Physical Performance among Community-dwelling Older Adults in a Community-based Falls Prevention Program
International Journal of Environmental Research and Public Health
(1) This study is a retrospective/secondary analysis of existing data from the CHAMP. This study should be improved in one of two ways because current one is not answering an important question.
In the introduction, just before the purpose statements, it is written that “Additionally, understanding characteristics of participants who are less likely to demonstrate improvements may assist providers with decision-making when considering individuals’ need for referral to other resources”; therefore, I expected that the study would find who did not improve. But the study purpose was to find who improved. The author should find out who did not improve in this study, only then the current conclusion can be stated. The conclusion was not directly delivered from this study. The statistical analysis should be the same for logistic regression, except the authors need to discuss who did not improve.
If that is not what the authors want to do, it should be stated upfront that the CHAMP used OEP in a community-setting and the purpose is to compare the OEP’s home-based results in the original study and CHAMP’s community-based results. In this way, the authors can use almost all parts of the manuscript except for the rational (the reasons why this study should be conducted) for the study.
- To address these recommendations, Lines 73-75 were deleted. In lines, 78-79, the following phrase was added, “and to compare these characteristics to those predicting improvement in the traditional home-based OEP.”
- Additionally, we updated the conclusion (see Lines 354-358).
The following suggestions may be still relevant after revising the major issue above.
(2) Add the number of data that was analyzed (128?) in the abstract.
- This information has been added in the abstract (see Line 22).
(3) Clarify the duration of Otago Exercise program per person (6 months?) in the abstract.
- The duration of the program has been added (see Lines 23-24).
(4) Explain the setting of assessment and Otago training and what instruction for home-based exercise was provided (community-setting?).
As recommended, we added additional description about the setting (see Lines 103-105) and instruction (see Lines 111-116).
(5) On page 4, line 158, CRT should be TUG. But the results of TUG should be categorized to improved and not-improved and use logistic regression.
- We appreciate this thorough review and your catching this error. We have corrected CRT corrected to TUG (now line 176).
- We chose to use linear regression versus logistic regression with TUG after evaluating the linear and categorical distributions of this variable. Further, we examined the univariate analyses with individual predictor variables. Based on these associations and TUG score distributions, we determined that linear regression was most appropriate for statistical modeling.
(6) Lines 264-268 is difficult to understand because of the double negative word “decreased characteristics of frailty.” The same request for the conclusion.
- We agree with the clarity of these referenced statements and have revised them to increase reader comprehension (see Lines 289 and Lines 354-358).
(7) Lines 264-274. Why this paragraph is related to hypothesis or research question (RQ)? If it is, state the related hypothesis or RQ.
- In response to this comment, we have deleted Lines 290-296.
(8) The rest is good. It all made sense when I read Lines 287-300, but what it means is that until that paragraph, I was not sure why the authors are not answering the important questions.
(9) Explain why OEP was adapted for a community-setting in CHAMP, when it was meant for home-based exercise intended for those whom tend to be home-bound? Older adults who can come to the community-setting is different from older adults who tend to be home-bound.
- An additional description was provided about the rationale for program adaptation on Lines 62-63.
(10) In CHAMP, explain what was provided for home OEP. No all readers know details of CHAMP.
- We have added an explanation about the CHAMP program on Lines 111-116.
Round 2
Reviewer 2 Report
My concerns were all addressed.